# Visualizing the Invisible: Visual-Based Design and Efficacy in Air Quality Messaging

**DOI:** 10.3390/ijerph182010882

**Published:** 2021-10-16

**Authors:** Zoey Rosen, Channing Bice, Stephanie Scott

**Affiliations:** Department of Journalism and Media Communication, Colorado State University, Fort Collins, CO 80523, USA; channing.bice@colostate.edu (C.B.); sms.scott@colostate.edu (S.S.)

**Keywords:** health communication, visual messaging, air quality, efficacy

## Abstract

This study examines the effect and efficacy of visual designs for messages about poor air quality. The study utilized a 2 (message efficacy: high vs. low) × 2 (message design: visual vs. text) between-subjects experimental design, of *N* = 95 students from a large Western university. This experiment assessed the effects of message design and efficacy of language on students’ visual comprehension, source credibility, self-efficacy, and protective behavioral intention. Hypotheses 1 and 2 were partially supported, finding that there were some statistically significant effects for efficacy and message design on students’ comprehension and protective behavioral intention. Future work should focus on strategies for more salient air quality health communication because wildfires will continue to impact the western United States.

## 1. Introduction

In the fall of 2020, western states in the United States experienced an unusually terrible fire season, lasting months. Between the effects of the Cameron Peak Fire and the East Troublesome Fire, the Rocky Mountain region was clouded in a 100-day haze [1]. Students unaccustomed to such fire events may not have known which protective measures to take for their health. Air quality risk is often communicated in the United States using the EPA’s Air Quality Index (AQI), which provides a categorization of risk that is dependent on the numerical concentration of air pollutants [2]. The AQI ranges from 0 to 500; the higher the number, the worse the air quality. When readings are higher than 100, there is an increased risk of poor health outcomes for sensitive groups, such as people with respiratory problems, the elderly, and small children. Readings between 201 and 500 signal to the public that the air quality is very unhealthy and can have negative health effects [2]. When there is an active wildfire, it is more likely that one will be able to see a haze in the air from the ground, but this is not always the case. Air quality is not always easy to “see” [3]; therefore, understanding the ways in which air quality can impact personal health is important. This research intends to emphasize that even healthy people can experience health impacts from polluted air. As such, understanding how to more effectively communicate simple precautions people can take to decrease the risks regarding exposure to wildfire smoke is important.

Although research has evaluated communication campaigns that address various health impacts of wildfires, previous efforts do not adequately address the communication of air quality mitigation behaviors before an active threat [4]. In a systematic analysis and through multi-sited interviews, Ramírez et al. (2019) found that there were three major themes regarding air pollution communications: (1) ambiguity around which information sources were responsible for communicating about air quality; (2) existing communication strategies did not contain sufficient vital information, such as risk mitigation behaviors or the long-term health impacts of air quality; and (3) existing communication strategies did not adequately reach vulnerable populations. Recognizing these three deficiencies in the researchers’ area, this study sought to create messaging strategies for air quality risk for the vulnerable population of undergraduate students. University students are some of the most vulnerable populations to natural disasters; they are less prepared to cope in an emergency, due to the transient nature of their residency in the community, lack of prior experience with regional natural hazards, and lower levels of self-responsibility [5,6]. In creating a message about how to mitigate the long-term health effects of poor air quality coming from their university, this study has the opportunity to experimentally test if a university is perceived as a credible information source during an environmental health threat, if there are more effective ways to communicate protective health behaviors, such as in visuals or through text, and if vulnerable populations such as college students can increase their personal self-efficacy around a hazard such as poor air quality.

This study investigated the influence of air quality message efficacy and design by utilizing a 2 (message efficacy: high versus low) × 2 (message design: visual or text-based) between-subjects experimental design. Guided by visual, cognitive, health, and risk communication literature, this study aims to elaborate on the relationships between message content and message design to create more salient protective health messages ahead of future poor air quality threats.

## 2. Literature Review

### 2.1. Visual Communication

#### 2.1.1. Working Memory, Comprehension, and Cognition

Visual representations have been proven to serve as powerful cognition tools. As images develop across scientific and non-scientific domains alike, visual information can both familiarize and de-familiarize our own personal perceptions and understandings of events [7,8,9,10]. Likewise, visual representations also help to enable learning and interpretations by promoting the connection of new ideas to pre-existing knowledge. Researchers suggest that well-crafted images have the ability to quickly and easily build upon abstract relationships between known and unknown concepts or elements. The ability to visualize those relationships enables simpler connections of observations to reasoning, which serves as a basis for externalized cognition [8,11,12,13]. Similarly, visualizations that encompass structures pertaining to knowledge and information have demonstrated the ability to help users understand complex topics and subjects that were not previously clearly defined [8]. This touches on the idea of “distributed cognition”, which recognizes that cognition, in itself, is computational, and as such, reaches across disciplines and utilizes visual models to help with the transportation of knowledge models across people or disciplines [7,14]. This field also explores relationships between visual and mental models to demonstrate how visual models assist in cognitive processing by developing analogies as they construct new models of knowledge.

In exploring these examples within the context of health communication, it is notable that even those with high literacy levels will commonly seek assistance when they encounter difficulties in processing complicated health information. In fact, patients tend to respond in a more positive manner to information when they experience higher levels of message comprehension [15]. Studies have demonstrated that incorporating visual messages alongside text has been shown to be more persuasive and more stimulating, in turn generating an emotional response in the reader and triggering memory systems within the brain to provide a rapid response [15]. This has also been shown to improve user attention, adherence, and perhaps most importantly, the recall of health information [15]. When used effectively, visual information also possesses the ability to reduce anxiety, increase feelings of self-efficacy, and generate a more confident mindset [12,16].

As such, research within the field suggests that health professionals can improve their communication on new technologies by incorporating visual materials into their message designs [17,18]. Visual information has the ability to enable and empower viewers by giving them a greater sense of control and confidence when using new technologies and, in turn, encouraging the more successful adoption of new health technologies into common health practices and behaviors [16]. The Cognitive Theory of Multimedia Learning (CTML) framework aligns the broad concepts of perception, cognitive processing, and interpretation, by illustrating the value of visuals for communicating technical information about new technologies. Mayer (2002) notes that CTML works by combining the cognitive processes of how people learn from presentations of instruction to promote more effective learning strategies. CTML posits that when a viewer receives information presented through both words and images via sensory mechanisms (e.g., eyes and ears), messages are processed through two separate channels, where the viewer actively selects pictures or words from their sensory memory and organizes the gathered information into working memory [19]. At this point, the information is processed as parts of pictorial or verbal models, which then become integrated with existing knowledge within the user’s long-term memory.

The Cognitive Theory of Multimedia Learning focuses on four major components from which people learn, which are as follows: dual-coding theory, limited capacity working memory, active processing, and information transfer. Through incorporating these concepts, Mayer (2002) developed the multimedia principle, which suggests that individuals learn better from both pictures and words, rather than words alone. Additionally, this principle further illustrates that using words and images when presenting information in multimedia contexts helps the cognitive processing of information and allows viewers to build stronger connections between pictorial and verbal mental constructs [19,20,21,22].

Based on visual communication literature applications to environmental health communication highlighted above, the following hypothesis is posed:

**Hypothesis** **1:***Individuals exposed to visually based air quality messages will experience greater levels of comprehension, protective behavioral intention, and perceived source credibility than those exposed to text-based air quality messages*.

#### 2.1.2. Social Cognitive Theory

Social Cognitive Theory outlines the two major factors that influence whether an individual will take preventative action: (1) if they believe that the benefits outweigh the costs of the behavior; and (2) if they believe they have the agency (self-efficacy) to perform the behavior [23,24]. The study of preventative behaviors is especially interesting within the cognition scholarship, because these actions are often undertaken whether or not the future event would occur. Individuals are more motivated to regulate their behaviors in the present when they anticipate a future state in which they could be affected by an event [25]. Health communication campaigns have studied this for many conditions, most infamously for anti-smoking campaigns to prevent lung cancer [26]. However, researchers have applied Social Cognitive Theory to natural hazard disaster preparedness, such as earthquake preparedness in the home [27]. Although it is not guaranteed that an earthquake will occur, having the enactive attainment, or experience, with preparedness behaviors for an earthquake was found to have a statistically significant, positive relationship with behavioral intention [27]. The behavioral intention to prepare for an earthquake was also found to be mediated by content self-efficacy [27]. Content self-efficacy describes the agency deriving from the knowledge that pertains to a particular situation [27]; for example, content self-efficacy has been measured using social cognitive theory and earthquake preparedness literature to more specifically describe efficacy in an earthquake-specific context.

#### 2.1.3. Efficacy

Protective behaviors and efficacy have also been researched within the field of risk communication. When risk messages include information about self and response efficacy, alongside pertinent hazard information, there is an increased likelihood of displaying self-protective behaviors [28]. From the risk perception scholarship that centered on self-protective behaviors, Kievik and Gutteling (2011) describe self-efficacy as one’s evaluation of their ability to undertake a protective action (e.g., “Am I able to protect myself from this risk successfully?”), whereas response efficacy describes one’s evaluation of the advised actions (e.g., “Is the advice I’m given about how to protect myself from this risk going to be successful?”) [28]. Both self-efficacy and response efficacy are required for an individual to undertake a preventative behavior [29]. Thus, because this study focused on the effects that efficacy has on preparedness behaviors for poor air quality in wildfire smoke events, both dimensions of efficacy will be measured to describe efficacy as a full construct.

Verroen, Gutteling, and de Vries (2013) conducted a 2 × 2 experiment comparing efficacy beliefs (which include self-efficacy and response efficacy) with peer feedback for the hazard of environmental exposure to ammonia fires and their adverse health effects. They found that participants who received articles with more efficacy information and also received supportive peer feedback via social networking site (SNS) messages were more likely to express higher levels of involvement and greater intentions to engage in protective behavior, as opposed to the low-efficacy condition [30]. The authors’ final recommendation for risk communication experts was that official hazard messaging should try to enhance levels of efficacy beliefs in future risk scenarios. Thus, this current study aimed to experimentally compare efficacious messaging with other constructs that have been shown to affect information processing within visual and crisis communication.

#### 2.1.4. Crisis and Risk Communication

The exchange of messages with the intent to avoid or decrease the negative outcomes of a crisis is known as crisis communication [31]. Fearn-Banks (2002) expands on that definition as, “verbal, visual, and/or written interaction between an organization and its stakeholders to initiate stakeholders (often through media) prior to, during, and after a negative occurrence” [32] (p. 480). Typically, these messages come from authority figures or emergency managers seeking to explain a particular event, its probable outcomes, and specific information on reducing harm to impacted communities in a transparent, timely, and accurate manner [33]. Risk communications, then, are messages with technical experts or scientists as a source that are more persuasive in nature with self-efficacy actions that can be taken to reduce negative outcomes [34,35,36]. Crisis and risk communications share several commonalities and intersect at a variety of points throughout a crisis while relying on the persuasive attribute of source credibility [37,38].

#### 2.1.5. Source Credibility

Source credibility has been identified as a construct within many studies to encapsulate the characteristics attached to a communicator that often has a persuasive influence, onto the audience receiving the message [39]. Over time, researchers have expanded on the conceptual definition of source credibility with different factors. Notably, Hovlan, Janice, and Kelley’s (1953) source credibility theory requires two constructs: expertise and trustworthiness. Expertise refers to the audience’s perception that the speaker possesses the ability to make correct assertions, and trustworthiness refers to the audience’s perception that the assertions made by the speaker are correct [40]. When both dimensions are taken into account, source credibility can be conceptually defined as the perception that the speaker of a message can be entrusted to make factual claims [40].

Universities are equally at risk of experiencing natural disasters as other institutions [41]. Universities have closed for extended periods of time and have been majorly damaged due to earthquakes, hurricanes, and tornadoes [41]. Watson, Loffredo, and McKee (2011) found that after forced evacuation due to Hurricane Ike, students wanted disaster preparedness to be covered during orientation, including evacuation checklists, how to access medical care when the university health center was closed, as well as instructions for what to do when university communications go down [42]. Universities, therefore, become important information sources during an emergency, where a student body looks to for instruction during a hazard [43]. Thus, measuring the perceptions of a university’s credibility could reveal an important mediating factor for their emergency and hazard messaging.

Based on the literature outlined above from the cognition and risk communication scholarship, the following hypothesis arises:

**Hypothesis** **2:**
*Individuals exposed to high-efficacy air quality messages will experience greater levels of comprehension, protective behavioral intention, and perceived source credibility than those exposed to the low-efficacy air quality message.*


Based on social cognitive, risk perception, and visual communication literature applications to environmental health communication, the authors sought to investigate if there was an interaction effect between independent variables. Therefore, the last hypothesis being posed is:

**Hypothesis** **3:**
*Message design and efficacy will interact so that individuals exposed to high-efficacy, visually based messages will demonstrate greater levels of comprehension, protective behavioral intention, and perceived source credibility than those exposed to the other message conditions.*


#### 2.1.6. Efficacy and Information Seeking Applications

The similarities and interconnectedness of crisis and risk communication prompted the Centers for Disease Control and Prevention (CDC) to develop the Crisis and Emergency Risk Communication (CERC) Model [38]. The creation of this stage model aimed to adapt the organizational definitions of crisis and risk communication to be more compatible with health communication during an era of emerging global public health threats, natural disasters, and bioterrorism. It recognizes that communication in this context must be prompt, strategic, and comprehensive [37,38]. Aerts (2013) applied the CERC model within the context of extreme weather by using a two-by-two between-subjects design to investigate the timing (pre-crisis vs. post-crisis) and efficacy beliefs (low vs. high). Results in this study revealed that it is significantly more likely that individuals will seek out information in the initial event and maintenance stages of a crisis event, compared to pre-crisis events [44].

In addition to the context of crisis and risk scenarios, information-seeking has been extensively studied within health communication. Anker, Reinhart, and Feeley (2011) performed a systematic literature review of health-information-seeking measures for 648 articles published between 1978 and 2010. Items falling under the theme of “general health information seeking” were the most popular, comprising 41.2% of the sample [45]. General health-information-seeking items measure how a participant engages in health information searches, usually on a dichotomous or Likert scale. Questions specifically ask about the source types that participants use to find information on a specific topic (e.g., searching the Center for Disease Control websites for information regarding the flu), as well as behavioral measures about the search process itself (e.g., number of calls to a health hotline). For this study, the dichotomous general health information scale was adapted for the topic of air quality in order to gain insights on this tendency by our participants. Especially as more people are actively searching for health information both on- and off-line [45], capturing these behaviors could illuminate how different populations engage with health information for hazards such as poor air quality.

Literature on health information-seeking behaviors for air quality specifically is lacking for the population of interest of the current study, prompting the following exploratory research question:

**RQ1:** 
*What information-seeking behaviors do college students practice for poor air quality?*


The previous hypotheses were statistically analyzed via this experiment in order to provide empirical evidence for the connections between concepts. The exploratory research question is addressed in the participants’ section and will be addressed in the discussion as well.

## 3. Methods

### 3.1. Procedure and Design

This study investigated the influence of self-efficacy and visual content by utilizing a 2 (message efficacy: high versus low) × 2 (message design: visual or text-based) between-subjects experimental design. Following the informed consent, participants were instructed to view a message on air quality and the health impacts of exposure to polluted air. Four air quality messages were created as stimulus material where participants were randomly assigned to one of four treatment groups: a high-efficacy text-based message, a high-efficacy visually based message, a low-efficacy text-based message, or a low-efficacy visually based message. Across all conditions, messages were formatted to look like an email message from their university with consistent background colors, headers, and browser bars. Experimental stimuli messages are shown in Figure A1, Figure A2, Figure A3 and Figure A4 in Appendix A. After the participant viewed one of the four randomly assigned conditions, participants moved to the post-test questionnaire, which aimed to capture the influence of message efficacy and format on visual comprehension, source credibility, and protective behavioral intentions. Lastly, participants’ demographic information was collected, which also included details on air quality information-seeking behavior and general health status.

### 3.2. Participants

Participants (*N* = 104) were recruited using the SONA research participant management system to compensate students enrolled in Department of Journalism and Media Communication classes with extra credit. To be eligible, participants were required to be college students and at least 18 years of age or older to provide informed consent. Data were cleaned for incomplete responses, which yielded a final sample of *N* = 95 where 23 participants viewed the high-efficacy text-based message, 25 participants viewed the high-efficacy visually based message; 23 participants viewed the low-efficacy text-based message, and 24 participants viewed the low-efficacy visually based message. Participants ranged in age from 18 to 34 years old (*M* = 21.03, *SD* = 2.62). The majority of participants identified as female (60%), with the remainder of participants identifying as male (38%) or non-binary (2%). Over two-thirds of the sample (67%) classified themselves as White, and the remainder identified as Asian (11%), Hispanic (10%), American Indian/Alaskan Native (3%), Black (3%), or preferred not to identify (3%). Most of the sample classified themselves as undergraduate sophomores (36%), with the remainder classifying themselves as undergraduate juniors (33%), seniors (28%), and freshmen (4%).

### 3.3. Manipulated Independent Variables

#### 3.3.1. Message Efficacy

For the first factor of message efficacy, participants were randomly assigned to conditions where they viewed a low- or high-efficacy message on precautions that can be taken to reduce the risk of adverse health outcomes from wildfire smoke exposure. The EPA’s Air Quality Index (AQI) information provided in the message included an explanation of the numerical value that the index displays, such as what a value of 100 means. Additionally, information about what populations are affected by the different numerical values in the index was written to further explain the potential impacts of poor air quality and those most at risk for each designation on the index. Low-efficacy messages included information about how the AQI is used to measure air quality conditions and detailed the symptoms people may experience as a result of wildfire smoke exposure. Similarly, the high-efficacy message also contained this information, but was manipulated to include the easy precautions one could take to decrease the risk of adverse health outcomes that result from wildfire smoke exposure. The design strategy for this variable was adapted from Verroen, Gutteling, and de Vries (2013) [30].

#### 3.3.2. Message Design

The second factor then embedded the low- or high-efficacy messages into either a text- or visually based message design. The text-based message only used text to convey air quality messaging while utilizing the university’s logos, branding colors, and fonts. Although the visually based message also used text, it utilized more colors from the university’s branding palette and integrated more visual depictions (e.g., AQI scale, iconography) to convey air quality information.

### 3.4. Dependent Variables

#### 3.4.1. Visual Comprehension

Nine Likert scale items (*M* = 3.71, *SD* = 0.71, α = 0.85) were used to measure visual comprehension. These items, adapted from extant literature on visual health information, demonstrated strong reliability for capturing participants’ preferences for learning about health risk information using visual communication strategies [46]. Participants indicated their level of agreement from strongly disagree (1) to strongly agree (5), with statements such as, “I often find that health risk information that uses words, but no pictures, is harder to follow” or “If I needed to change my behavior due to a health risk, a visual illustration of the steps I could take to lessen the threat would help me better understand the risk”.

#### 3.4.2. Protective Behavioral Intent

Using six Likert scale items that demonstrated strong reliability (*M* = 4.11, *SD* = 0.71, α = 0.81) measured the participant’s protective behavioral intentions in response to the air quality message. Participants reported their level of agreement from not at all helpful (1) to extremely helpful (5) that the listed behaviors would help in protecting them from poor air quality events. Examples of these behaviors included statements such as, “Close the windows in your home, even if it is hot outside” or “Running a HEPA-certified air filter in your home”.

#### 3.4.3. Source Credibility

Eight seven-point semantic differential items (*M* = 6.14, *SD* = 0.74, α = 0.90) demonstrated strong reliability in measuring perceived source credibility adapted from Ohanian (1990) [39]. These items captured how participants perceived the credibility of the university as the source of the air quality message. Example anchors for these items included, “undependable/dependable”, “insincere/sincere”, and “unqualified/qualified”.

## 4. Results

### 4.1. Manipulation Check

To ensure that the stimulus messages were being perceived as low- or high-efficacy, participants were asked six Likert scale items (*M* = 4.26, *SD* = 0.63, α = 0.83) adapted from Aerts (2013) [44]. Participants reported their level of agreement from strongly disagree (1) to strongly agree (5) with statements such as, “If I take the recommended preparation for poor air quality, I run less risk” or “I can perform the recommended preparation to prevent me from experiencing health risks from poor air quality”. As anticipated, an independent-samples t-test revealed that those in the high-efficacy condition (*M* = 4.41, *SD* = 0.08) perceived their self-efficacy to be significantly, *t*(93) = −2.46, *p* = 0.02, higher when exposed to the high-efficacy message, than the low-efficacy message (*M* = 4.10, *SD* = 0.65). A statistical manipulation check was not run for the text-based versus visual-based design condition because the researchers determined that, based on the previous literature around visual designs and the resulting created stimuli, the manipulation was obvious and did not need to be measured beforehand.

### 4.2. Findings

To analyze all hypotheses, two-way ANOVA comparisons were used. Table 1 displays a table of the means and standard deviations for all experimental groups, whereas Table 2 displays a table of all the ANOVA results.

Hypothesis 1 predicted that those exposed to the visually based message design would experience greater levels of comprehension, protective behavioral intention, and perceived source credibility than those exposed to the text-based message. There was a significant main effect of message design on visual comprehension scores, where participants exposed to the visually based message (*M* = 3.92, *SD* = 0.64) reported significantly greater comprehension, *F*(1, 91) = 9.48, *p* < 0.01 than those exposed to the text-based message (*M* = 3.48, *SD* = 0.75). Contrary to expectations, participants exposed to the text-based message design (*M* = 4.24, *SD* = 0.63) reported higher levels of protective behavioral intent than those exposed to the visually based message design (*M* = 3.99, *SD* = 0.77); however, no significant differences were found, *F*(1, 91) = 3.14, *p* = 0.08. Additionally, participants exposed to the visually based message design (*M* = 6.16, *SD* = 0.78) reported slightly higher levels of perceived source credibility than those exposed to the text-based message design (*M* = 6.12, *SD* = 0.71), with no significant differences *F*(1, 91) = 0.05, *p* = 0.82. The aforementioned findings partially support hypothesis 1.

Hypothesis 2 predicted that those exposed to the high-efficacy message would experience greater levels of comprehension, protective behavioral intention, and perceived source credibility than those exposed to the low-efficacy condition. Although participants exposed to the high-efficacy message (*M* = 3.76, *SD* = 0.71) reported higher levels of visual comprehension than those exposed to the low-efficacy message (*M* = 3.65, *SD* = 0.76), no significant differences were detected. *F*(1, 91) = 0.06, *p* = 0.44. As anticipated, there was a significant main effect of message efficacy on protective behavioral intent scores, where participants exposed to the high-efficacy message (*M* = 4.27, *SD* = 0.71) reported significantly greater protective behavioral intent, *F*(1, 91) = 5.21, *p* = 0.03, than those exposed to the low-efficacy message (*M* = 3.94, *SD* = 0.69). Additionally, those exposed to the high-efficacy message (*M* = 6.22, *SD* = 0.73) also reported higher perceived source credibility than those in the low-efficacy message condition (*M* = 6.07, *SD* = 0.75); however no significant differences were found *F*(1, 91) = 0.89, *p* = 0.35. Thus, Hypothesis 2 was found to be partially supported.

Hypothesis 3 predicted that the message design and message efficacy will interact so that participants exposed to high-efficacy, visually based messages will demonstrate greater levels of comprehension, protective behavioral intention, and perceived source credibility. Although those exposed to the high-efficacy visually based message reported the highest comprehension scores (*M* = 3.95, *SD* = 0.64) of those exposed to the other message conditions, there was not a statistically significant interaction between the effects of message efficacy and design on visual comprehension scores, *F*(1, 91) = 0.11, *p* = 0.74. Conversely, those exposed to the high-efficacy text-based message reported the highest protective behavioral intent scores (*M* = 4.38, *SD* = 0.70) of those exposed to the other message conditions; however, there was not a statistically significant interaction between the effects of efficacy and message design on protective behavioral intention, *F*(1, 91) = 0.04, *p* = 0.84. Additionally, those exposed to the high-efficacy visually based message reported the greatest perceived source credibility scores (*M* = 6.31, *SD* = 0.69) of those exposed to the other message conditions. There was not a statistically significant interaction between the effects of efficacy and message design on source credibility, *F*(1, 91) = 1.10, *p* = 0.30. These results indicate that Hypothesis 3 was not supported.

The exploratory research question (RQ1) asked about the information-seeking behaviors college students practiced for poor air quality and general health status. Nearly half of the sample (43%) reported that they do seek out air quality information during poor air quality events, with the remainder reporting that they sometimes (38%), or do not (19%) seek more information. Those participants who reported that they did, or sometimes did seek out more air quality information were instructed to select all sources they visited during poor air quality events (*n* = 77). The majority of participants indicated that they most often turn to the National Weather Service (71%). The remainder reported seeking out more air quality information from state and local public health authorities (62%); family and friends (55%); local news media outlets (47%) such as TV, radio, or newspapers; social media (45%); on- or off-campus healthcare providers (39%); air quality websites such as Purple Air and EPA AirNow (31%); and other sources (3%) such as Google or mobile weather applications. The majority of participants reported their general health status as being very good (53%), in concordance with previous literature on the health status of college student populations [47]. The remainder of the participants classified their health as being good (27%), excellent (15%), or fair (4%), with no one reporting a poor health status.

About half of the sample (43%) reported that they do seek out air quality information during poor air quality events, with the remainder reporting that they sometimes (38%), or do not (19%) seek more information. Interestingly, the majority of participants indicated that they most often turn to the National Weather Service for air quality information (71%), followed by state and local public health authorities (62%), then family and friends (55%), and then healthcare providers (39%).

## 5. Discussion

The current study contributes to air quality messaging literature that has not often been studied, although it is becoming more salient with each wildfire season. Through a 2 × 2 experimental design, this study’s independent variables of message efficacy (high vs. low) and design (text vs. visual) were investigated. Hypothesis 1 predicted that the visual condition will have an effect on comprehension, source credibility, and protective behavioral intention, as opposed to the text-based condition. There was a significant main effect of message design on visual comprehension scores where participants exposed to the visually based message reported significantly greater comprehension than those exposed to the text-based message. This agrees with the literature around message comprehension and visual-based design [16,21]. However, there were not statistically significant effects for the main effects of visual design on protective behavioral intention or perceived source credibility; thus, hypothesis 1 was only partially supported. Interestingly, we expected that those exposed to the visually based message would report higher levels of protective behavioral intention because of the increased comprehension, but this was not the case in our study. The statistically non-significant relationship between protective behavioral intention and the message format could potentially be explained by other artifacts, such as issue salience. The study was conducted between the end of the previous fire season and months before the start of the next season; therefore, participants may not have had as high motivation to engage in protective behavioral measures because there was no active threat. Additionally, the non-significant finding around source credibility could potentially speak towards channel effects. The choice of the message design in an email format from the university may have had an unintentional effect; students experience high rates of message fatigue from institutional emails, especially around health messages [48]; therefore, participants may have perceived the source credibility differently, regardless of the text-based or visual-based messaging format.

Hypothesis 2 predicted that the high-efficacy condition will have an effect on comprehension, protective behavioral intention, and source credibility compared to those exposed to the low-efficacy condition. As anticipated, there was a significant main effect of message efficacy on protective behavioral intent scores where participants exposed to the high-efficacy message reported significantly greater protective behavioral intention than those exposed to the low-efficacy message. High efficacy within a message has been shown to increase protective behavioral intention; thus, our findings are in line with previous work in this area [30]. However, hypothesis 2 was only partially supported, because there were statistically non-significant findings for the effect of the high-efficacy condition on visual comprehension or perceived source credibility than those in the low-efficacy message condition. This may be attributed to the smaller sample size, which will be discussed in the limitations section below. Another possible explanation for the non-significant findings around high-efficacy language and message comprehension could be due to the subject of air quality information itself. In the United States, the comprehension of air quality varies by previous experience with wildfire smoke, health status, risk perceptions, and various other factors [49]. In this study, more information about access to exposure-reducing resources from the university as well as from students’ previous hometowns could elaborate on the baseline comprehension this sample has for air quality information.

The partial support of Hypotheses 1 and 2 indicates that further research is needed to investigate how text- or visually based, high- and low-efficacy messages impact comprehension, source credibility, and protective behavioral intention. Self-efficacy for environmental hazards could be a promising avenue of research, especially because preparedness behaviors have implications on health and safety. Especially because the salience of understanding and engaging with air quality health information is becoming more important with each fire season, these findings are essential as a foundation for messaging strategies. Hypothesis 3 predicted that there will be an interaction effect between the visual-based and high-efficacy condition on comprehension, protective behavioral intention, and source credibility. Although there was not a statistically significant interaction effect between the two independent variables, high-efficacy and visual-based design, previous studies have shown that these experimental manipulations should have an impact on audience perceptions of the messages’ content and source [27,34]. The effect efficacy and visual message design have on these perceptions has been found to guide behavioral intentions relative to the presented health and safety information [27,34]. The mean scores for participants that had the high-efficacy condition and visual-based messages followed the predictions from past literature, but were non-significant at the *p* < 0.05 level. A replication of this study with more participants in each condition could confirm the hypothesized trends with fewer constraints regarding statistical power.

The exploratory research question (RQ1) asked about the information-seeking behaviors college students practiced for poor air quality. Information-seeking on air quality specifically for this population is vastly under-researched; therefore, the exploratory findings were encouraging. About half of the sample (43%) reported that they do seek out air quality information during poor air quality events. Regarding what sources students turned to for this air quality information, the majority referred most often turn to the National Weather Service (71%), followed by state and local public health authorities (62%), then family and friends (55%), and then healthcare providers (39%). The sources were not mutually exclusive, meaning students could select more than one source from the list. These findings somewhat support the previous literature on the health-information-seeking behaviors of college students, which suggests that students are most likely to often or always use the internet for health information, followed by healthcare providers, and friends or family [50]. In the air quality context, students reported similar patterns in information-seeking; digital sites such as the National Weather Service website were used most often, followed by public health authorities, and then friends and family. Seeking information from experts in online spaces and in their personal networks within the domain of air quality is an encouraging finding. By referencing credible sources, such as the NWS or state and local agencies, students are displaying where they believe they can access information about the different hazards they experience. Gaining a better understanding of where students search for information about air quality can be used by universities in order to boost their message credibility around the health risks during wildfire season and can be important for community stakeholders in preparation for mitigation messaging, building community partnerships, and other pre-season strategies.

### Limitations and Future Work

The statistically non-significant results attributing to the partial support of the hypotheses may be a result of the study’s smaller sample size. Although student participants were incentivized with extra credit, the ongoing demands of the COVID-19 pandemic during the study timeframe may have been a deterrent for opting to perform more tasks online in addition to the required coursework. Additionally, this study was designed to be relevant for the recent wildfires that happened within 15 miles of the university campus. Although the debriefing statement and informed consent documents provided access to mental health resources and the ability to opt out of the experiment, participants could have been discouraged from initial participation because the material may have triggered traumatic memories. Additionally, the results found could have limits in their external validity due to the salience of the issue for the region.

Furthermore, there may be a limitation in the generalizability of this student sample because the students were all enrolled in a course in the Department of Journalism and Media Communication. Those that are majoring in Journalism may be more attuned to message transmission from a variety of mediated channels, which could also account for the lack of significant differences. However, most of the students in the sample pool were enrolled in a common-core technical writing course, which includes majors from the entire university; therefore, the singular department recruitment may not have been as large of a limitation. Especially because the context for visual-based design and efficacy in air quality messaging is more novel, future work or iterations of similar experiments should include more data collection in the quantity of participation and quantity of university departments to better capture these variables’ effects. Individual differences, e.g., learning preferences for text or images, could also be added as additional dependent variables in future iterations of this study [51].

## 6. Conclusions

This study experimentally examined the effect of efficacy and visual design for messaging for air quality. Air quality is not always easy to “see”; therefore, understanding the health risks from exposure to wildfire smoke, as well as precautions that can be taken to decrease these risks, is important. There is a paucity around the communication of air quality mitigation behaviors before an active threat [4]. Universities have been looked to as credible sources of information for their students experiencing natural disasters; therefore, it is important that the improved message strategies are prioritized for routine hazards such as poor air quality during wildfire season, especially because students are a vulnerable population [5,6]. Self-efficacy and visual design should be further investigated in environmental health communication contexts, because visual information has been proven to reduce anxiety, increase feelings of self-efficacy, and generate a more confident mindset when used effectively [12,16]. This study’s hypotheses were partially supported, finding that there were some statistically significant effects for efficacy and message design on comprehension and protective behavioral intention. Practical implications that can be drawn from the findings of this study are that high-efficacy messaging is very important when the goal of the message is to influence behavior, and when combined with other persuasive appeals, has the potential to be very effective in environmental health messaging. Air quality messaging construction and design will become all the more critical in the coming years to protect people from the adverse health effects of wildfire smoke.

## Figures and Tables

**Table 1 ijerph-18-10882-t001:** Descriptive statistics.

			Mean	Standard Deviation	N
Comprehension	Low Efficacy	Text	3.40	0.79	23
		Visual	3.90	0.65	24
		Total	3.65	0.76	47
	High Efficacy	Text	3.56	0.73	23
		Visual	3.95	0.64	25
		Total	3.76	0.71	48
	Total	Text	3.48	0.75	46
		Visual	3.92	0.64	49
		Total	3.71	0.73	95
Behavior	Low Efficacy	Text	4.09	0.53	23
		Visual	3.81	0.80	24
		Total	3.94	0.69	47
	High Efficacy	Text	4.38	0.70	23
		Visual	4.16	0.71	25
		Total	4.27	0.71	48
	Total	Text	4.24	0.63	46
		Visual	3.99	0.77	49
		Total	4.11	0.71	95
Credibility	Low Efficacy	Text	6.13	0.64	23
		Visual	6.01	0.85	24
		Total	6.07	0.75	47
	High Efficacy	Text	6.11	0.78	23
		Visual	6.31	0.69	25
		Total	6.22	0.73	48
	Total	Text	6.12	0.71	46
		Visual	6.16	0.78	49
		Total	6.14	0.74	95

**Table 2 ijerph-18-10882-t002:** Table of two-way ANOVA results.

		*df*	Mean Square	*F* Value	*p*
Comprehension	Message design	1	4.68	9.47	0.00 **
	Message efficacy	1	0.29	0.59	0.44
	Design * efficacy	1	0.06	0.11	0.74
	Within groups (error)	91	0.49	-	-
Behavior	Message design	1	1.52	3.14	0.80
	Message efficacy	1	2.52	5.21	0.03 *
	Design * efficacy	1	0.20	0.04	0.84
	Within groups (error)	91	0.48	-	-
Credibility	Message design	1	0.30	0.05	0.82
	Message efficacy	1	0.49	0.89	0.35
	Design * efficacy	1	0.61	1.10	0.30
	Within groups (error)	91	0.56	-	-

Note: * denotes significance at *p* < 0.05 and ** denotes significance at *p* < 0.01.

## Data Availability

The data presented in this study are available on request from the corresponding author. The data are not publicly available due to secure storage method approved by Institutional Review Board.

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
