# Peer review of "Visualizing the Invisible: Visual-Based Design and Efficacy in Air Quality Messaging"

_ijerph, 2021, doi:10.3390/ijerph182010882_

Round 1
Reviewer 1 Report
This study experimentally examined the effect of efficacy and visual design for messaging for air quality in the student environment. The paper is well structured and easy to read. However, the abstract must be improved. It should better summarize the main massage of the paper. The study analyses a rather small sample. Can we draw mayor conclusions based on 95 questioners? Moreover, the authors are using parametrical statistical tests on a non-continuous dataset. In this case, non-parametrical tests should be applied. I recommend revising the results and resubmitting the paper.
Additional comments are in the PDF document.

Reviewer 2 Report
This paper is important given the current concerns with wildfires and air contamination in the West Coast of the US. It is well-designed and written up. There are some issues that need to be clarified before the paper can be published:
1) The statements in the first paragraph of the Introduction need to be supported by references.
2) At the end of Section 2.1.2, the term "content self-efficacy" needs to be defined.
3) In Section 2.1.3, it states that, "Self-efficacy describes one's evaluation of their ability to undertake a protective action." This definition is not an accurate definition of self-efficacy, which is defined as one's belief in one's ability to produce certain outcomes based on one's confidence in one's abilities. Is there a term such as "protective self-efficacy" to describe this type of self-efficacy?
4) In Section 3.2.1, the content of the Air Quality Index should be explained.
5) The information in the second paragraph of Section 3.4 seems redundant with that which is presented at the end of the Results section. This information should only be presented in the Results section since it answers the authors' posed research question.
6) The first three paragraphs of the Discussion section describe how Hypotheses 1 and 2 were partially supported and how Hypothesis 3 was not supported. Each paragraph should contain a more thorough explanation of the non-significant findings than simply small sample size, and there should be rationales given for each non-significant result.
7) The discussion in the Discussion section of the National Weather Service being the most used source for air quality information was confusing. The authors state that seeking scientific experts and public health authorities online was encouraging; the National Weather Service does not seem to fit into either of those two categories.
8) The authors did not mention the limitation of all students being from the Department of Journalism and Media Communication. There are generalizability limitations from only using one college department. Also, students in this major may be particularly attuned to how messages are transmitted or used to interpreting media messages in a wide variety of formats, which could account for the lack of significant differences. This limitation should be mentioned.
9) On the last page in the Conclusion, the authors state that there were statistically significant effects on comprehension, source credibility, and protective behavioral intention. However, there were no statistically significant effects on source credibility.
Round 2
Reviewer 1 Report
The authors improved the text in regard to reviewer's comments. Almost all sugestions were implemented and explanations added. The methodological part is now better presented and supported with proper citations. I recommend publishing the paper.
Reviewer 2 Report
All of this reviewer's concerns were adequately addressed.